# Management of Hyperthyroidism during Pregnancy: A Systematic Literature Review

**DOI:** 10.3390/jcm12051811

**Published:** 2023-02-24

**Authors:** Aida Petca, Daiana Anne-Marie Dimcea, Mihai Cristian Dumitrașcu, Florica Șandru, Claudia Mehedințu, Răzvan-Cosmin Petca

**Affiliations:** 1Department of Obstetrics and Gynecology, “Carol Davila” University of Medicine and Pharmacy, 050474 Bucharest, Romania; 2Department of Obstetrics and Gynecology, Elias University Hospital, 011461 Bucharest, Romania; 3Department of Obstetrics and Gynecology, University Emergency Hospital, 050098 Bucharest, Romania; 4Department of Dermatology, “Carol Davila” University of Medicine and Pharmacy, 050474 Bucharest, Romania; 5Department of Dermatology, Elias University Emergency Hospital, 011461 Bucharest, Romania; 6Department of Obstetrics and Gynecology, Filantropia Clinical Hospital, 011171 Bucharest, Romania; 7Department of Urology, “Carol Davila” University of Medicine and Pharmacy, 050474 Bucharest, Romania; 8Department of Urology, “Prof. Dr. Th. Burghele” Clinical Hospital, 050659 Bucharest, Romania

**Keywords:** hyperthyroidism, pregnancy, thyroid function, thyroid disease

## Abstract

In pregnancy, several physiological changes affect maternal circulating thyroid hormone levels. The most common causes of hyperthyroidism in pregnancy are Graves’ disease and hCG-mediated hyperthyroidism. Therefore, evaluating and managing thyroid dysfunction in women during pregnancy should ensure favorable maternal and fetal outcomes. Currently, there is no consensus regarding an optimal method to treat hyperthyroidism in pregnancy. The term “hyperthyroidism in pregnancy” was searched in the PubMed and Google Scholar databases to identify relevant articles published between 1 January 2010 and 31 December 2021. All of the resulting abstracts that met the inclusion period were evaluated. Antithyroid drugs are the main therapeutic form administered in pregnant women. Treatment initiation aims to achieve a subclinical hyperthyroidism state, and a multidisciplinary approach can facilitate this process. Other treatment options, such as radioactive iodine therapy, are contraindicated during pregnancy, and thyroidectomy should be limited to severe non-responsive thyroid dysfunction pregnant patients. In light of this events, even in the absence of guidelines certifying screening, it is recommended that all pregnant and childbearing women should be screened for thyroid conditions.

## 1. Introduction

Hyperthyroidism is a rare condition that affects about 0.1–0.4% of all pregnancies, characterized by high values of circulating thyroid hormones (T4 and T3) as well as a low value of TSH [1,2].

The most common causes of hyperthyroidism in pregnancy are Graves disease and hCG-mediated hyperthyroidism (for example, gestational transient hyperthyroidism, hyperemesis gravidarum, and gestational trophoblastic disease). Other less common causes include toxic thyroid adenoma, subacute thyroiditis, and drug-induced hyperthyroidism [3,4,5].

The symptoms of hyperthyroidism in pregnancy include heat intolerance, anxiety, fatigue, tachycardia, tremor, nausea alongside vomiting, and weight loss despite retained appetite [3,4,5]. Other manifestations, such as confusion or increased metabolic rate, appear as severe complications, known as a thyrotoxic crisis, that can be induced by labor, cesarean section, or gestational trophoblastic disease [5,6].

Pregnant women with hyperthyroidism require careful evaluation because they have an increased risk of miscarriage, preeclampsia, preterm birth, and heart failure [7]. Because of its adverse effects on both the mother and the fetus, the rapid recognition and proper management of cases are imperative to preventing major complications [8,9].

Antithyroid drugs are the most common therapy carried out in pregnant women with hyperthyroidism, as surgical treatment (thyroidectomy) can be grafted with complications and iodine therapy is contraindicated in pregnancy [9,10,11].

Currently, there is no consensus regarding an optimal method to treat hyperthyroidism in pregnancy. Thus, the study’s main objective is to review the specific considerations, findings, and new approaches used in the management of hyperthyroidism during pregnancy.

## 2. Materials and Methods

The term ’hyperthyroidism in pregnancy’ was searched in the PubMed and Google Scholar databases to identify relevant articles published between 1 January 2010 and 31 December 2021. All resulting abstracts that met the inclusion period were evaluated. The materials and methods sections from all included articles were assessed to identify details related to hyperthyroidism-complicated pregnancies and iodine intake.

The inclusion criteria were:Literature published in the English language during the study period (1 January 2010–31 December 2021);Publications that corresponded to the term search (“hyperthyroidism in pregnancy”);Publications which included all causes of hyperthyroidism alongside pregnancy;Publications which included diagnosed hyperthyroidism before or during pregnancy;Publications which included management of hyperthyroidism complications;Publications which included management of iodine deficiency in pregnancy.

A study was not included if:Only the abstract was accessible;Published aside from the mentioned period (1 January 2010–31 December 2021);Not published in the English language;Did not use the mentioned topic;Was identified as a preclinical study;Has been restricted to the pediatric or neonatal population.

The term “hyperthyroidism in pregnancy” was searched on the PubMed and Google Scholar databases. Filters were applied and included: articles in English, human studies, full texts, and the mentioned publishing period. From Google Scholar, duplicate articles and citation links of other previously included articles were excluded.

A total of 858 studies from both databases were identified. All resulting studies were screened based on the title and abstract, and 409 studies were excluded. The majority of the excluded studies did not fall into the targeted theme (hyperthyroidism management in pregnancy) or targeted population. After full text screening, the most relevant 57 articles were selected to form the final review—Figure 1.

## 3. Changes in Thyroid Function during Pregnancy

The thyroid gland undergoes physiological changes from the beginning of pregnancy, both by increasing in size and vascularity. Human chorionic gonadotropin beta (β-HCG) stimulates the thyroid gland from the first trimester due to its thyroid-stimulating hormone (TSH)-like structure. The thyrotropic action of β-HCG also causes a decrease in the value of TSH in the first trimester of pregnancy, explaining why pregnant women have lower values of TSH than non-pregnant patients [12]. The maximum level of hCG is reached at 8–10 weeks of gestation, then decreases and remains in a plateau until term; the TSH value reaches the minimum point at approximately the same time [13,14]. In twin pregnancies, hCG growth is more pronounced and is responsible for thyroid stimulation, leading to more frequent and higher values of fT4 and TSH suppression [14].

The resultant thyroid hormone changes in pregnancy also come from the increased concentration of maternal estrogen, which induces a substantial raise in the circulating level of thyroxine binding globulin (TBG), stepping up thyroxine (T4) by up to 50%. Higher TBG levels increase the ability to bind T4 and thus reduce the free form of T4 (fT4) (Figure 2) [13]. The underlying mechanism consists of an increased rate of TBG synthesis by hepatocytes and the reduction of its plasma clearance [14]. Thyroxine (T4) and triiodothyronine (T3) concentrations increase in the first weeks of pregnancy and reach a plateau in the second trimester, having concentrations that are 30–100% higher than preconceptionally [12].

Another important event that affects maternal thyroid function during pregnancy is a change in thyroid hormones’ peripheral metabolism. These variations occur in pregnancy due to the process of deiodination at the placental level [16]. The uteroplacental deiodination enzymes predominant form is D3, which prevents the activation of T4 and inactivates T3, protecting the fetus from excessive exposure to maternal thyroid hormones. The minority form of placental deiodinase, D2, is considered to be especially important early in pregnancy in maintaining the intraplacental level of T3 required for trophoblast development [14,16].

Another change in thyroid function that occurs during pregnancy appears as a re-duction in maternal iodine deposits as a result of three events: increased iodine intake required for maternal thyroxine synthesis; increased clearance; transfer of iodine from mother to fetus (Table 1) [14].

## 4. Thyroid Hormones Trimester-Specific Reference Range

Multiple studies have described the reference range for thyroid hormone levels depending on each trimester. However, the range varies greatly depending on ethnicity, the amount of sample, the iodine concentration, and the laboratory method used [17,18].

TSH is minimal in the first trimester of pregnancy and increases thereafter. According to the latest published studies, both the American Thyroid Association (ATA) and the Endocrinology Society suggest that the upper limit for TSH is 2.5 mlU/L in the first trimester, 3.0 mlU/L in the second trimester, and 3.5 mlU/L in the third. The lower limit is 0.1 mlU/L in the first trimester, 0.2 mlU/L in the second trimester, and 0.3 mlU/L in the third trimester [16,17,19]. It is difficult to determine a lower limit of TSH normal range, especially in the first trimester of pregnancy. As mentioned before, the correlation with hCG reduces TSH concentration in serum, which makes the diagnosis of early hyperthyroidism in pregnancy difficult [20]. According to recently published studies, the lower limit for TSH was different depending on the characteristics of the studied groups [20]. In a study conducted by Khalid et al. in Ireland, the lower limit of the TSH reference interval was 0.1 mlU/L [21]. In contrast, in another study conducted in the Czech Republic by Springer et al., the lower limit according to the criteria applied to the group of patients was set at 0.25 mlU/L [22].

Considering the TSH trimester-specific reference interval in pregnant women, the incorrect application of the standard interval for a non-pregnant patient leads to an overestimation of the number of patients classified as ‘hyperthyroid.’ Judging by the physiological changes mentioned, it is not uncommon for a healthy pregnant woman to present with suppressed TSH and fT4 at the upper limit of normal. This situation is relatively common with the physiological changes that occur in the first trimester, is not associated with side effects, and does not require treatment [23].

Thyroxine is considered to be an important element as it is used to diagnose thyroidian pathology. Currently, its free form, fT4, is used in the laboratory to determine hypo/ hyperthyroidism. To avoid the possibility of false-positive results, the American Thyroid Association (ATA) guidelines support the implementation of a TSH and fT4 trimester-specific reference range by using a single dosing method [24]. According to included studies, during the first trimester, there is a slight increase in fT4 serum concentration, which can no longer be included in the reference range of a non-pregnant patient [20]. In the next two trimesters, there is a decrease in fT4 serum concentration due to the thyroid response produced by the increase in free thyroxine-binding protein, an increase in the fetal need for thyroid hormone synthesis in the first trimester, and metabolic changes due to placental formation. The lower fT4 serum level in the third trimester is a normal process in healthy pregnant women which does not affect pregnancy and fetal development [25].

Therefore, according to the recommendations of the Endocrinology Society, TSH, fT4, and, less frequently, fT3 levels should be determined using trimester-specific reference range intervals [20].

## 5. Thyroid Screening in Pregnancy

Thyroid pathology is the second-most common endocrine condition that affects fertile women. If left untreated in pregnancy, it is associated with an increased risk of miscarriage, abruptio placentae, pregnancy-induced hypertension, and intrauterine growth restriction [26]. Hyperthyroidism occurs in approximately 0.1–0.4% of pregnancies and is characterized as fT4 or fT3 serum levels, or both, above the trimester-specific reference range. The most common cause of hyperthyroidism in pregnancy is Graves’ Disease. Less common forms are transient pregnancy thyrotoxicosis or a multinodular goiter [27].

Currently, the Endocrinology Society recommends thyroid screening only in high-risk pregnant women (Table 2) [16,17,26]. However, in 2012, in publishing new guidelines, the Endocrinology Society board could not reach a consensus on the recommendations for testing thyroid function in all pregnant patients. Some board members recommended thyroid screening for all pregnant women in the ninth week of gestation. In contrast, others recommend thyroid screening only for high-risk patients; however, in reality, every physician should decide for their patients [27].

In the case of patients who have already been treated for hyperthyroidism prior to pregnancy with synthetic antithyroid drug therapy, surgery, or radioiodine therapy, and who are currently euthyroid (whether or not they are receiving thyroxine), neonatal hyperthyroidism may still occur [7].

In pregnant women with active Graves disease, TSH receptor antibodies (TRAbs) should be determined at the time of presentation to the physician to assess the severity and risk of fetal hyperthyroidism. Such conduct is indicated for all patients with a history of surgery or radioiodine therapy. A new reassessment is mandatory between 22 and 26 weeks of gestation. If the serum level exceeds three times the upper limit, rigorous monitoring of the fetus through ultrasonography is necessary [28,29]. In the case of elevated TSH receptor antibody (TRAb) values at 36 weeks of gestation, the newborn should be evaluated postpartum for hyperthyroidism [29]. In some cases of pre-pregnancy hyperthyroidism, it may remit during pregnancy; however, it will reappear postpartum when the immune status returns to the Th1 (T-cell helper type 1) state [7].

## 6. Hyperthyroidism Treatment in Pregnancy

Antithyroid drugs are the main therapeutic form used in pregnant women. Radioactive iodine treatment is contraindicated in pregnancy, and surgery should be reserved for severe conditions that do not respond to treatment with antithyroid drugs [2,30]. Antithyroid drugs inhibit thyroid hormone synthesis by reducing iodine organification and coupling monoiodothyrosine with diiodothyrosine [2].

The antithyroid drugs that are most commonly used are methimazole (MMI), propylthiouracil (PTU), and carbimazole (CMZ). Propylthiouracil should be limited to the first trimester, after which it will be replaced with methimazole because it risks inducing congenital disabilities (dextrocardia, dysgenesis/renal agenesis) as well as an increased risk of liver toxicity [2,29,31,32].

However, some studies do not describe the occurrence of birth defects in pregnant women with hyperthyroidism treated with antithyroid drugs. In a survey of 2.830 pregnant women with hyperthyroidism, the birth defects rate was similar in the antithyroid drugs treated group and the control group [28,33].

The main goal of the treatment is to maintain the fT4 value at the upper limit of the trimester-specific reference range by using the lowest possible dose to prevent maternal and fetal complications (intrauterine growth restriction, fetal tachycardia, fetal heart failure, fetal hydrops, accelerated bone maturation) [19,34]. Although there are side effects of antithyroid drugs which present in pregnant women treated with antithyroid drugs, the risk of maternal and fetal complications in untreated patients is much higher.

The recommended loading dose for propilthyouracil is 50–150 mg orally three times per day, depending on the severity of the symptoms. When methimazole is used, the recommended loading dose is 10–40 mg orally three times per day [29,35]. At the beginning of treatment, pregnant women with hyperthyroidism should be monitored every 2–4 weeks for optimal dosage [36].

All antithyroid synthesis drugs cross the placental barrier and can cause inhibition of the fetal thyroid. Propylthiouracil (PTU) is more soluble in water and is found attached to the binding globulin in a higher concentration compared to methimazole, which suggests that methimazole induces a higher transplacental passage. PTU is recommended during pregnancy to prevent the risk of developing fetal hypothyroidism [36].

Another therapeutic class used for a limited period is beta-blockers. Still, recent studies do not recommend them for long periods, as they may cause intrauterine retard, and, if administered antepartum, they can induce neonatal transient hypoglycemia or bradycardia [2].

The current treatment options are limited to controlling thyroid hormone secretion. The main concern in recent studies is the uncertainty in choosing the optimal antithyroid drug [28,37,38]. Until recently, the recommended treatment for hyperthyroidism in pregnancy was PTU, followed by MMI and CMZ; however, recent studies have suggested a risk of liver dysfunction attributed to PTU. The US FDA and the National Institute of Child Health and Development from the USA have changed their guidelines, recommending the limitation of PTU use to the first trimester of pregnancy [16,29].

## 7. Maternal and Fetal Adverse Events of Untreated/Uncontrolled Hyperthyroidism during Pregnancy

Rapid intervention in hyperthyroidism during pregnancy prevents both maternal and fetal complications. The absence of intervention or low control in the management of hyperthyroidism in pregnancy is associated with abortion, pregnancy-induced hypertension, preeclampsia, low birth weight, intrauterine growth restriction, and thyrotoxicosis [30]. Interestingly, in the study conducted by Turunen et al., obstetrical complications were demonstrated more frequently in women diagnosed with hyperthyroidism prior to pregnancy, suggesting that such a diagnosis may represent an additional residual risk of complications [39].

Preeclampsia affects up to 8% of pregnancies worldwide and is a leading cause of maternal mortality and morbidity [40]. Maternal hyperthyroidism has been associated with an increased risk of developing pregnancy-induced hypertension. The risk of developing preeclampsia was shown to be lower in patients who did not use antithyroid medical therapy during pregnancy, suggesting that the risk was higher for patients with an active form [39]. Thyroid hormones play an important role in placental development and are an important regulator of inflammatory and metabolic processes [41]. In a study carried out in the Netherlands in 2014, a 3.4 times higher risk of developing pregnancy-induced hypertension was demonstrated compared to the group of euthyroid patients [42]. In another study conducted by Saki F. et al. in 2014, the prevalence of preeclampsia in pregnant women with hyperthyroidism was 6% compared to 0% in a euthyroid group and 7.5% in pregnant women with hypothyroidism [43]. However, hypothyroidism is the main cause of preeclampsia from thyroid disorders [23,44,45]. For the correct management of the disease, it is necessary to update diagnostic techniques to prevent maternal and fetal complications. In this vein, Korevaar et al. demonstrated that additional hCG measurements can be used to identify pregnant women with increased susceptibility to developing thyroid pathologies within the second trimester, this study being the first one which demonstrated this hypothesis. This higher risk appears to be mediated by sinergic effects of autonomous thyroid hormone synthesis or TSH receptor antibodies and hCG stimulation synthesis, which produce increased thyroid hormone secretion [41].

The pathogenic mechanism of hyperthyroidism is involved in the occurence of obstetric complications in pregnant women with this condition [30]. Multicenter studies and case-control studies included in our review demonstrate that hyperthyroidism in pregnancy can be associated with an important fetal impact, such as intrauterine growth restriction (IUGR) and being small for its gestational age [46,47,48].

Intrauterine growth restriction (IUGR) is an important marker of intrauterine fetal development which has a particular impact throughout the entire life of the child. This term is used to describe a fetus that does not reach the optimal growth curve, diagnosis of which is possible with fetal biometry or Doppler velocimetry. The definition of intrauterine growth restriction is fetal weight below the 10th percentile for gestational age, for which a frequently encountered maternal factor is thyroid dysfunction [49,50]. Patients with uncontrolled hyperthyroidism have a 9.2 times greater risk of having small for gestational age newborns compared to pregnant women without thyroid dysfunction. Even in controlled hyperthyroidism, the risk of intrauterine growth restriction increased 2.3 times [51].

An important factor for normal intrauterine development is also represented by fetal sex. In a study led by Aiken C.E. et al. suggested and then demonstrated that male fetuses grew faster than female fetuses and were more likely to be macrosomic, while female fetuses were more likely to develop intrauterine growth restriction [50,52]. Furthermore, according to the study, even the placenta is affected differently depending on the fetal sex: a female fetus placenta is more resistant to insults compared to a male fetus placenta, sustaining the different impact of maternal thyroid hormones on both sexes [52]. Furthermore, in intrauterine growth restriction pregnancies, a certain pattern was identified: terminal villi reduce trophoblast proliferation and angiogenesis with increased matrix proteins deposits in villous stroma [52].

Abruptio placentae (placental abruption) remains a severe and rare obstetrical complication, with a frequency between 0.6% and 1% in the United States of America and 0.4–0.5% in European countries. It consists of chronic placental dysfunction as well as the early separation of the placenta from the uterine wall before the onset of labor, as well as, which, through progression, decreases the surface area used for fetal oxygen exchange. This process leads to a small for gestational age newborn, prematurity, and perinatal mortality. In a severe form, it progresses rapidly with a large volume of maternal blood loss, fetal hypoxia, and intrauterine death requiring rapid diagnosis and intervention [53]. In a meta-analysis performed in China covering the literature from 2010 to 2021 that evaluated the effect of thyroid dysfunction in terms of obstetric outcomes, a considerable heterogeneity in the incidence of placental abruption between the normal group and thyroid dysfunction group (OR = 0.27, 95% CI: 0.19–0.38, Z = 7.52, *p* < 00001) was demonstrated. The incidence of abruptio placentae in the group with thyroid dysfunction was significantly increased compared to the control group (*p* < 0.05) [54]. According to one of the studies included in this meta-analysis, there is a significant statistical correlation between hyperthyroidism, placental abruption, labor induction, and advanced maternal age [55]. Thus, correcting hyperthyroidism during pregnancy reduces the risk of this complication [53]. Turunen et al. did not identify any differences in risk of placental abruption between patients with hyperthyroidism undergoing treatment and the euthyroid group [39].

## 8. Iodine Intake in Pregnancy

During pregnancy, the iodine intake must increase by 50% due to the needs of the maternal body to synthesize more thyroid hormones, raised renal excretion due to increasing the rate of glomerular filtration, and enhanced fetal requirements from the second trimester [19].

The US Institute of Medicine (IOM) and the World Health Organization (WHO) have recommended a daily intake of 220–250 mcg of iodine for pregnant women. According to most studies, the recommended method to determine iodine consumption is by measuring the urinary iodine concentration, with typical values ranging from 150–249 µg/L [56,57].

A study conducted in Israel in 2017 that aimed to determine urinary iodine concentration, including in pregnant women, showed iodine deficiency in this group to be over 85%, with an average of 61 µg/L (IQR 36–97 µg/L) [58].

Thus, to ensure that this vulnerable patient category has a sufficient level of iodine in organism, various national and international organizations recommend iodine supplementation in pregnant women [16,59].

However, in 2007, the World Health Organization (WHO) published a document attesting that, in the case of pregnant women living in regions with adequate iodine intake, defined as a median urinary iodine concentration (UIC) >100 µg/L for more than two years and the use of iodized salt in more than 90% of homes, supplementation is not necessary, pregnant women being directly protected by using iodized salt [59,60]. An additional source of Iodine is the use of animal products. In the United Kingdom, the addition of iodine products to animal diet and the use of iodine supplements have led to increased iodine levels in products of animal origin. The accidental contribution of an increased iodine dose through these methods contributes to reducing iodine deficiency internationally and is considered an ’accidental triumph of public health sector’ [59]. Thus, to prevent iodine deficiency in the United Kingdom, it is recommended that pregnant women should consume two–three portions of dairy products per day to achieve their daily iodine requirements; a glass of milk is considered to provide 50 µg of iodine, which is equivalent to 20% of the recommended intake for pregnant women [61].

According to the latest American Thyroid Association guidelines, pregnant women with urinary iodine concentrations ranging from 50 to150 µg/L fall into the mild to moderate iodine deficiency category [57,62].

In national studies including childbearing-age women from different European countries, the average iodine intake was about half of the recommended World Health Organization (WHO)’s recommended level. The optimal daily iodine intake during pregnancy recommended by the World Health Organization (WHO) and International Council for Control of Iodine Deficiency Disorders (ICCIDD) is 200 µg/L and for the US Institute of Medicine (IOM) it is 220 µg/L; in contrast, in Germany, according to the study by Verbundstudie Ernahrungserhebung und Risikofaktoren-Analytik (VERA), the average iodine intake is 100 µg/L (33–284 µg/L) in women aged 19–24 [63]. In a study conducted in Denmark by Rasmussen et al., the average required intake for women aged 18 to 22 years old in Copenhagen was found to be 116 µg/L per day, while in the Netherlands, the National Food Consumption Survey has established a median intake of 149 µg/L per day for women aged 20–49 years old [63].

The consequences of severe iodine deficiency are well known and include fetal hypothyroidism and delayed neurological development [56]. Based on observational studies, it has been suggested that mild to moderate iodine deficiency, even in the presence of adequate maternal thyroid function, may be associated with impaired fetal neurodevelopment, characterized by a mildly lower IQ ranging to severe impairment of writing or reading [64,65]. The central role of maternal iodine in fetal development was demonstrated in the Avon Longitudinal Study of Parents and Children [64]. Studying children from monitored urinary iodine concentration mothers showed that an iodine/ creatinine ratio <150 µg/L was related to scores in the lowest quartiles for verbal IQ, reading, and the ability to understand a text [64]. These data indicate the importance of adequate iodine intake during pregnancy and highlight the risk of iodine deficiency even in developed countries [19].

Iodine deficiency-induced diseases in pregnancy appear when the pregnant woman suffers from reduced iodine intake, which leads to a reduction in thyroid hormone synthesis, affecting the muscles, heart, liver, and neurological development of the newborn [66]. The impact of iodine deficiency in pregnancy differs depending on the degree of severity and trimester range, this division being necessary in order to guide therapeutic strategies [67]. During the first trimester, glomerular filtration and blood flow rise, resulting in increased urinary iodine excretion. As an adaptive mechanism, the body raises the secretion of human chorionic gonadotropin (HCG) and thyroid-stimulating hormone (TSH), which leads to an increase in fT4 concentration that helps thyroid hormones synthesis [66]. Even mild iodine deficiency is a risk factor for diabetes mellitus and abruptio placentae. In a study conducted in China in 2018, maternal iodine deficiency induced an increase in serum thyroid-stimulating hormone (TSH) levels which caused an antagonist effect on insulin, generating hyperglycemia associated with increased oxidative stress, which is a risk factor for abruptio placentae [67].

Many authors suggest that iodine supplementation should be introduced in the first trimester of pregnancy to guarantee thyroid hormones synthesis and to avoid adverse maternal and fetal impacts during pregnancy [66,67]. Even if, in the second pregnancy trimester, the thyroid-stimulating hormone (TSH) level returns to normal, fetal development becomes indirectly dependent on maternal thyroid, since the fetal thyroid, although immature, begins hormone synthesis [66]. In case of mild iodine deficiency diagnosed in the second trimester of pregnancy, it is still possible to prevent complications through iodine supplementation [68,69]. In a study conducted in Thailand involving second trimester pregnant women, it was shown that iodine deficiency was further accentuated in this trimester, which increased the risk of preeclampsia, intrauterine growth restriction, and low birth weight [68]. In the third trimester of pregnancy, thyroid hormone-dependent neurogenesis is severe and irreversible, and is affected in the case of iodine deficiency that is present from the first trimester. Children born to mothers with such a deficiency may be small for gestational age (SGA) due to intrauterine growth restriction and reduced thyroid hormone production [66]. Iodine has an important antioxidant function; iodine deficiency increases the oxidative stress level, which can cause pregnancy induced hypertension [70]. The main target of iodine deficiency in pregnancy is the thyroid. Due to the autoregulatory mechanism, there is an increased thyroid stimulating hormone (TSH) secretion that causes thyroid volume increase and nodule development [66].

Despite the high prevalence of iodine deficiency in pregnant women, there is no exact correlation with maternal thyroid hormone serum levels [57]. However, a Belgian study on pregnant women showed an increase in thyroid disease prevalence in iodine deficiency patients. Even so, a significant association between urinary iodine concentration and TSH serum level could not be demonstrated, only a poor correlation with fT3 and fT4 [71].

Consequently, according to the new evidence, multiple organizations, such as the American Thyroid Association (ATA), Endocrinology Society (ES), and WHO, recommend using iodine supplementation during pregnancy. As a general rule, pregnant women with a median urinary iodine concentration of 100 µg / L or above do not require iodine supplementation during pregnancy [72]. In the European countries where iodine deficiency predominates, clinicians should consider recommending a pregnancy supplement with a daily iodine dose of 150 µg/L in pregnant women or in patients planning a pregnancy. Organizations such as the European Thyroid Association (ETA) and the European Office of International Council on Control for Iodine Deficiency Disorders (ICCIDD) should encourage more pregnancy supplement manufacturers to include optimal iodine doses (approximately 150 µg/L per day) [63].

## 9. Future Directions

The management of hyperthyroidism in childbearing women should begin in the preconception period to prevent maternal and fetal negative outcomes of this affected category in accordance with a multidisciplinary team including an endocrinologist, gynecologist, and neonatologist.

The therapeutic management process for pregnant women with hyperthyroidism presents certain issues of significant interest at the present time that require careful consideration. It is well known that hyperthyroidism management during pregnancy is challenging and can result in abnormal fluctuations in thyroid hormone levels. In this sense, according to the current knowledge, it is necessary to reassess the thyroid hormone trimester-specific reference range that is considered safe for the fetus on a regular basis, since some of the studies have demonstrated that even a value at the upper limits of the normal range can negatively impact the fetus’ development. Furthermore, further studies within this field are needed to create standardized protocols for the follow-up of these patients, and up until then, we recommend that all pregnant and childbearing women be screened for thyroid conditions even in the absence of guidelines certifying screening.

## 10. Conclusions

Hyperthyroidism is a leading cause of maternal morbidity during pregnancy which is associated with health risks for the fetus, as well. Thus, pregnant women with thyroid disease require careful observation and management in order to ensure a favorable outcome for the mother and newborn.

Currently, the main method used to maintain subclinical hyperthyroidism, validated by studies, is the use of antithyroid drugs; however, other therapeutic classes, such as beta-blockers, can be used for a limited period. Except for in medical therapy, radioactive iodine therapy is contraindicated, and surgical thyroidectomy is associated with both maternal and fetal complications and should be limited only to severe non-responsive medical treatment cases. It is clear that iodine intake plays an important role in pregnancy and should be increased by 50% to enhance fetal requirements, especially from the second trimester.

## Figures and Tables

**Figure 1 jcm-12-01811-f001:**
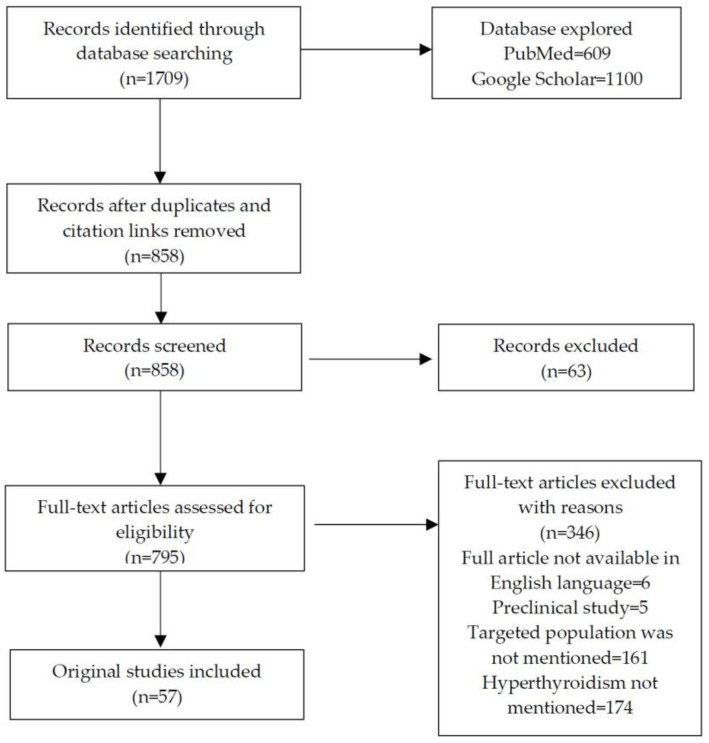
Literature selection and data extraction.

**Figure 2 jcm-12-01811-f002:**
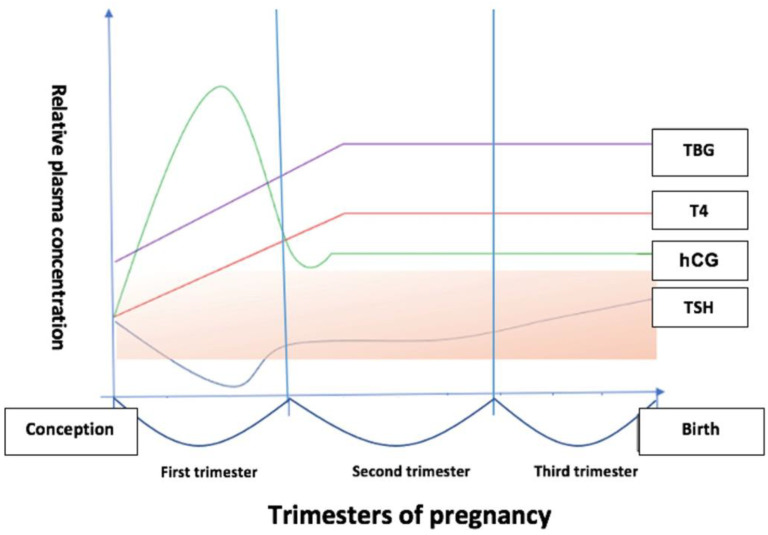
Changes in the plasma concentration of thyroid hormones and hCG in pregnancy. The orange surface corresponds to the normal reference range for non-pregnant women (hCG—human chorionic gonadotrophin; TBG—thyroid-binding globulin; T4—thyroxine; TSH—thyroid-stimulating hormone) [15].

**Table 1 jcm-12-01811-t001:** Physiological changes that affect maternal thyroid function in pregnancy [14].

Physiological Changes	Effect
1. Thyroid stimulation by hCG produced in trophoblastic tissue	Transiently increased fT4 and T3Decreased TSH levels
2. Estrogen-induced in-crease in serum TBG	Increased total T4 and T3
3. Placental expression of D2 and D3 deiodinases	Increased peripheral iodothyronine metabolism
4. (a) Increased renal Iode clearance; (b) Iode diversion to feto-placentar unit; (c) Increased iodide consumption for thyroid hormone synthesis	Reduced maternal iodide pool

**Table 2 jcm-12-01811-t002:** Endocrinology Society recommendations for thyroid screening in pregnancy [16,17].

Current Thyroid Therapy
Family history of autoimmune thyroid disease
Goiter
Personal background of: − autoimmune disease; − throat irradiation; − postpartum thyroid dysfunction; − newborn with thyroid disease; − hyperthyroidism treatment; − type 1 diabetes.

## Data Availability

All information included in this review is documented by relevant references.

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
