# Peer review of "Management of Hyperthyroidism during Pregnancy: A Systematic Literature Review"

_jcm, 2023, doi:10.3390/jcm12051811_

Round 1

Reviewer 1 Report

I congratulate the authors for a well written article. Please elaborate if there are any guidelines for screening of iodine deficiency in pregnancy.

Reviewer 2 Report

This review was compact and well written, and aim is important. The references are not very fresh, which may indicate that there are no new publications concerning the subject. Therefore, I think that in this form this review has no new data to offer for the reader.

However, in this article hyperthyroidism is well described as well as its risks and complications. Especially it has been focused to the risks of active disease during pregnancy. Less is known of the significance of prior hyperthyroidism to the pregnancy and its outcome. This has been studied lately (e.g. Turunen et al. Maternal hyperthyroidism and pregnancy outcomes: A population-based cohort study) and I think this kind of viewpoint would be interesting and important.

I also doubt that there is not yet sufficient grounds to recommend universal screening of thyroid conditions.

Round 2

Reviewer 2 Report

Thank you for considering my short comments. However, in this new form of article I find this extensive chapter of iodine intake somehow confusing under the current title of the article. I think there is no need to focus to this subject in this extent. I also think that the data to support these findings - e.g. iodine deficient as risk factor for abruptio placentae - is still quite weak. So I suggest to condense this addition.
